# Trace amines produced by skin bacteria accelerate wound healing in mice

Arif Luqman[1,2,3], Muhammad Zainul Muttaqin[3,4], Sumah Yulaipi[2], Patrick Ebner[1], Miki Matsuo[1], Susanne Zabel[5], Paula Maria Tribelli[1,6,7], Kay Nieselt [4], Dewi Hidayati[2✉] & Friedrich Götz [1✉]

Certain skin bacteria are able to convert aromatic amino acids (AAA) into trace amines (TA) that act as neuromodulators. Since the human skin and sweat contain a comparatively high content of AAA one can expect that such bacteria are able to produce TA on our skin. Here we show that TA-producing *Staphylococcus epidermidis* strains expressing SadA are predominant on human skin and that TA accelerate wound healing. In wounded skin, keratinocytes produce epinephrine (EPI) that leads to cell motility inhibition by β2-adrenergic receptor (β2-AR) activation thus delay wound healing. As β2-AR antagonists, TA and dopamine (DOP) abrogate the effect of EPI thus accelerating wound healing both in vitro and in a mouse model. In the mouse model, the *S. epidermidis* wild type strain accelerates wound healing compared to its Δ*sadA* mutant. Our study demonstrates that TA-producing *S. epidermidis* strains present on our skin might be beneficial for wound healing.

[1] Microbial Genetics, Interfaculty Institute of Microbiology and Infection Medicine Tübingen (IMIT), University of Tübingen, D-72076 Tübingen, Germany. [2] Biology Department, Institut Teknologi Sepuluh Nopember, 60111 Surabaya, Indonesia. [3] Generasi Biologi Indonesia (Genbinesia) Foundation, 61171 Gresik, Indonesia. [4] Aquaculture Department, Universitas Muhammadiyah Gresik, 61121 Gresik, Indonesia. [5] Center for Bioinformatics Tübingen, University of Tübingen, Sand 14, D-72076 Tübingen, Germany. [6] Departamento de Química Biológica, FCEyN-UBA, Buenos Aires, Argentina. [7] IQUIBICEN, CONICET, Buenos Aires, Argentina. ✉email: dewi_hidayati@ymail.com; friedrich.goetz@uni-tuebingen.de

Trace amines (TA) and classical neurotransmitter (dopamine, norepinephrine, or serotonin) are stored and released together in and from the nerve terminals[1,2]. TA play crucial roles physiologically as neuromodulators of synaptic transmission of classical neurotransmitter in mammalian brain by potentiating their activity[3,4]. Beside acting as neuromodulator, TA are reported to interact with a new family of G protein-coupled receptors (GPCRs), the TA-associated receptors (TAARs), thus they act independently from classical neurotransmitter[5,6]. Borrowsky and colleagues reported that phenylethylamine (PEA) and tyramine (TYM) modulate the synaptic excitation of dopaminergic neurons or alter the reactivity of dopamine D2 receptor (D2R) to ligands by activating TAAR1[5,7]. It has also been reported that TA interacts with various adrenergic receptors (ARs). TA act as agonists on α2-adrenergic receptor (α2-AR)[8–10] and as partial allosteric antagonists on β2-adrenergic receptor (β2-AR)[11].

Recently, it has been shown that various species of the genus Staphylococcus are able to produce TAs by using an enzyme called SadA (staphylococcal aromatic amino acid (AAA) decarboxylase)[8]. SadA is highly promiscuous because it not only decarboxylates all biogenic AAAs into tryptamine (TRY), PEA, and TYM but also dihydroxy phenylalanine (L-DOPA) and 5-hydroxytryptophan (5-HTP) to the neurotransmitter dopamine (DOP) and serotonin (SER).

First, it was not clear what might be the advantage to the bacteria of having SadA. The AAAs present in the medium are imported into the cell, pyridoxal phosphate (PLP)-dependent decarboxylated, and quantitatively excreted into the medium as TA[8]. There is no evidence that AAAs are used for protein biosynthesis or are involved in other metabolic pathways. Since energy-consuming reactions rarely occur in nature without a reason, the question arises as to what is the benefit of TA synthesis to the bacteria. As many TA-producing staphylococcal species are described as animal pathogens and skin colonizers, comparative studies with Staphylococcus pseudintermedius and its sadA deletion mutant were carried out. S. pseudintermedius is responsible for severe and necrotizing infections in humans and dogs[12–14]. Staphylococci are part of the human intestinal microflora and the TA-producing staphylococci are predominant among the staphylococcal genus present[8]. One reason for the predominance of TA-producing staphylococci in the gut could be that TA increase the internalization of bacteria into the host cells[8]. It is well documented that an increase in internalization of the epithelial cells protects the bacteria from both the host immune system[15–17] and antibiotics[18,19].

The molecular mechanism of how these neurochemicals boost bacterial internalization has been unraveled only recently[20]. TA ad DOP activate α2-AR and induce a reduction of the cytoplasmic cAMP level as well as an increased F-actin formation. This causes an increased internalization of the bacteria by the host cells. This internalization mechanism is independent from the FnBP-Fn-α5β1 integrin-mediated pathway[20]. The TA-triggered internalization of bacteria into the host cells represents a new invasion mechanism. The additional role of TA-producing bacteria play in the gut is unknown and such role is difficult to verify. However, in the meantime it emerged that neurotransmitters and neuromodulators are not only produced by the host, but also by the gut microbiota and that they play a role in the Gut-Brain axis[21].

Here we show that TA produced by skin bacteria accelerate wound healing by antagonizing the EPI-induced cell migration inhibition.

## Results

**TA-producing Staphylococcus epidermidis strains are predominant on human skin.** Since TA-producing staphylococci have an advantage in host cell adherence and internalization[8], we investigated whether and how many TA-producing staphylococci colonize the human skin and its prevalence. We took skin swabs from the forearm (antecubital fossa) of 28 healthy volunteers as illustrated in Supplementary Fig. 1. The swabs were examined for the presence of staphylococci using SK-salt agar, a selective medium for isolating coagulase-positive and coagulase-negative staphylococci[22]. All volunteers were colonized with staphylococci with the degree of colonization varied from 5 to 6000 CFU/100 cm² (Fig. 1a). Twenty out of 28 volunteers were colonized by TA-producing staphylococci (70%) (Fig. 1b). In total we isolated 900 colonies and tested them for TA production by

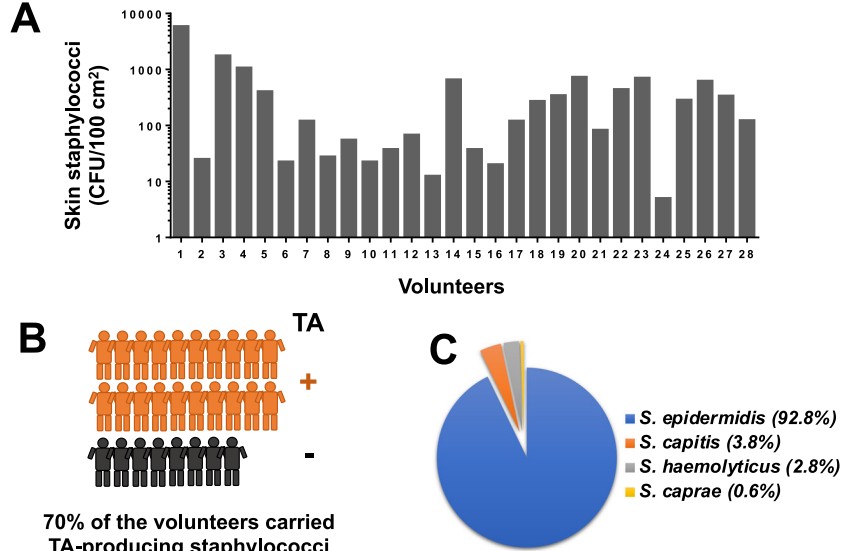

**Fig. 1 TA-producing S. epidermidis is prevalent on human skin. a** Number of staphylococci (CFU/100 cm²) found on the forearm (antecubital fossa) of 28 volunteers. **b** 900 random colonies from all volunteers were assayed by HPLC analysis for TA production: 70% of the volunteers carried TA producing staphylococci. **c** Of the 900 colonies, 185 (20%) produced TA; these colonies were identified at species level using 16s rRNA sequencing. The majority of TA producer belonged to S. epidermidis, followed by S. capitis, S. haemolyticus, and S. caprae.

**Table 1 Harvested staphylococcal colonies per volunteer and distribution of TA producers.**

| CFU range of staphylococci on volunteers' skin (CFU/100 cm²) | Harvested colonies analyzed for TA production by HPLC | Volunteers | Total analyzed colonies by HPLC (900) | TA producing strains |
|---|---|---|---|---|
| >1000 | 120 | 1, 3, 4 | 360 | **185 = 20%** |
| 500–1000 | 60 | 14, 20, 26, 23 | 240 | |
| 200–500 | 30 | 5, 18, 19, 22, 25, 27 | 180 | |
| 50–200 | 15 | 7, 9, 12, 17, 21, 28 | 90 | |
| 0–50 | ≤7 | 2, 6, 8, 10, 11, 13, 15, 16, 23 | 30 | |

**Table 2 Presence and identity of SadA in S. epidermidis.**

| S. epidermis strains | SadA | identity in % |
|---|---|---|
| O47 | + | 100 |
| ATCC 12228 | + | 100 |
| BPH0662 | + | 100 |
| DAR1907 | + | 100 |
| PM221 | + | 99,8 |
| CSF41498 | + | 99,8 |
| SEI | + | 99,8 |
| NCTC13924 | + | 99,6 |
| FDAARGOS_529 | + | 99,6 |
| CDC120 | + | 99,6 |
| CDC121 | + | 99,6 |
| ATCC 14990 | − | - |
| NBRC 100911 | − | - |
| RP62A | − | - |
| 14.1.R1 | − | - |
| FDAARGOS_153 | − | - |
| FDAARGOS_161 | − | - |
| 1457 | − | - |
| NCTC4133 | − | - |
| S. pseudintermedius ED99 | + | 52 |

HPLC analysis (Supplementary Fig. 2). Of the 900 staphylococcal colonies/strains 185 produced TA (20%) as shown in Table 1. We identified all of 185 TA producer using 16S rRNA sequencing. The majority of the TA-producing staphylococci belonged to S. epidermidis (92,8%) followed by S. capitis, S. haemolyticus, and S. caprae (Fig. 1c).

We also asked how prevalent the sadA gene is in the S. epidermidis species and blasted (tblastn of NCBI's Microbial BLAST) the SadA protein sequence of ATCC 12228 against 19 complete S. epidermidis genomes. 11 of the genomes (58%) encoded a highly homologous (99–100% identity) SadA protein, while other staphylococcus species, such as of S. pseudintermedius ED99 shared only 52% identity (Table 2).

**TA and DOP abrogate the EPI-induced cell migration arrest—thus accelerating the wound healing.** Given that human skin cells, particularly keratinocytes, express β2-adrenergic receptor (β2-AR)[23–26], whereas TYM and PEA are allosteric antagonists for β2-AR[11], we asked whether the TA produced by sadA-expressing staphylococci promote skin wound healing.

To address this question, we first carried out an in vitro migration assay with HaCaT cells, a human keratinocyte cell line that expresses β2-AR[27]. It is well known that the stress hormone epinephrine (EPI) acts as an agonist of the β2-AR, which lowers the migratory rate of keratinocytes and thus impairing wound healing[28,29]. We wanted to find out if TA and DOP, as agonists of the alpha2-adrenergic receptor (α2-AR)[20], are able to abrogate the EPI effect. The wound healing assay was carried out in the presence of EPI (25 μg/ml) alone and in combination with TA

(TRY, PEA, TYM) and DOP (each at 25 μg/ml). We included DOP in the experiment since the sadA-expressing staphylococci can also convert L-DOPA into DOP[8]. As a further control we included ICI 118551 (ICI), a selective β2-AR antagonist, also referred to as beta blocker[30]. For better comparison we used for all compounds at the same concentration of 25 μg/ml. At this concentration the effect of EPI on gap closing in HaCaT cells was most pronounced (Supplementary Fig. 3a).

The in vitro wound healing assay, which is mainly based on the determination of the host cell migration rate, was performed with HaCaT cells in culture-insert (2-wells format). In comparison to the control (untreated cells), EPI visibly delayed gap closing by lowering the migration of HaCaT cells. However, in the presence of TRY, PEA, TYM, DOP or ICI, the negative effect of EPI was abrogated and gap closing was almost as efficient as the control (Fig. 2a). The positive effect of ICI was mainly due to its inverse agonist effect on ß-AR[31,32]. The gap closing analyses using ImageJ showed that TA and DOP abrogated ($p < 0.05$) the negative effect of EPI on gap closing (Fig. 2b). In the absence of EPI, no effect on gap closing was seen with TA, DOP and ICI in comparison to the control (Fig. 2c). We also tested alprenolol (ALP, a neutral ß2-AR), which showed similar effect as ICI (Fig. S3b). Phentolamine (PTL) showed no effect in gap closing suggesting that α-AR does not play a role in gap closing of HaCaT cells (Supplementary Fig. 3b).

**TA and DOP enhance the migration of HaCaT cells even in the presence of mitomycin C.** Wound healing involves two distinct mechanisms, cell proliferation and cell migration. Therefore, we investigated cell migration and cell proliferation independently. Cell migration assay was carried out with HaCaT cells as described earlier, with the only difference that we added mitomycin C (10 μg/ml) to inhibit the cell proliferation.

In the absence of mitomycin C the gap closing was about 85% after 24 h (Fig. 2b, control). However, in the presence of mitomycin C, however the gap closing in the control decreased to only 15% after 24 h (Fig. 3a). In the presence of EPI gap closing decreased further to only 5%, while the addition of TA, DOP, and ICI abrogated the negative effect of EPI (Fig. 3a). In the presence of mitomycin C and the absence of EPI, TA, and DOP showed a similar gap closing as observed in the control (Fig. 3a, second half). In the presence of ICI, the gap closing was always higher. Here we show that the inhibition of cell proliferation by mitomycin C decreased the gap closing but had no principal effect on the activities of TA, and DOP, indicating that TA and DOP only affect cell migration but not proliferation.

**TA and DOP counteract the EPI-induced activation of β2-AR.** Activation of the β-AR by EPI inhibits the wound closing by inhibiting the migration of keratinocytes[29,33–35]. Since the activation of β-AR is followed by an increase in the intracellular cAMP level which decreases cell migration, we investigated

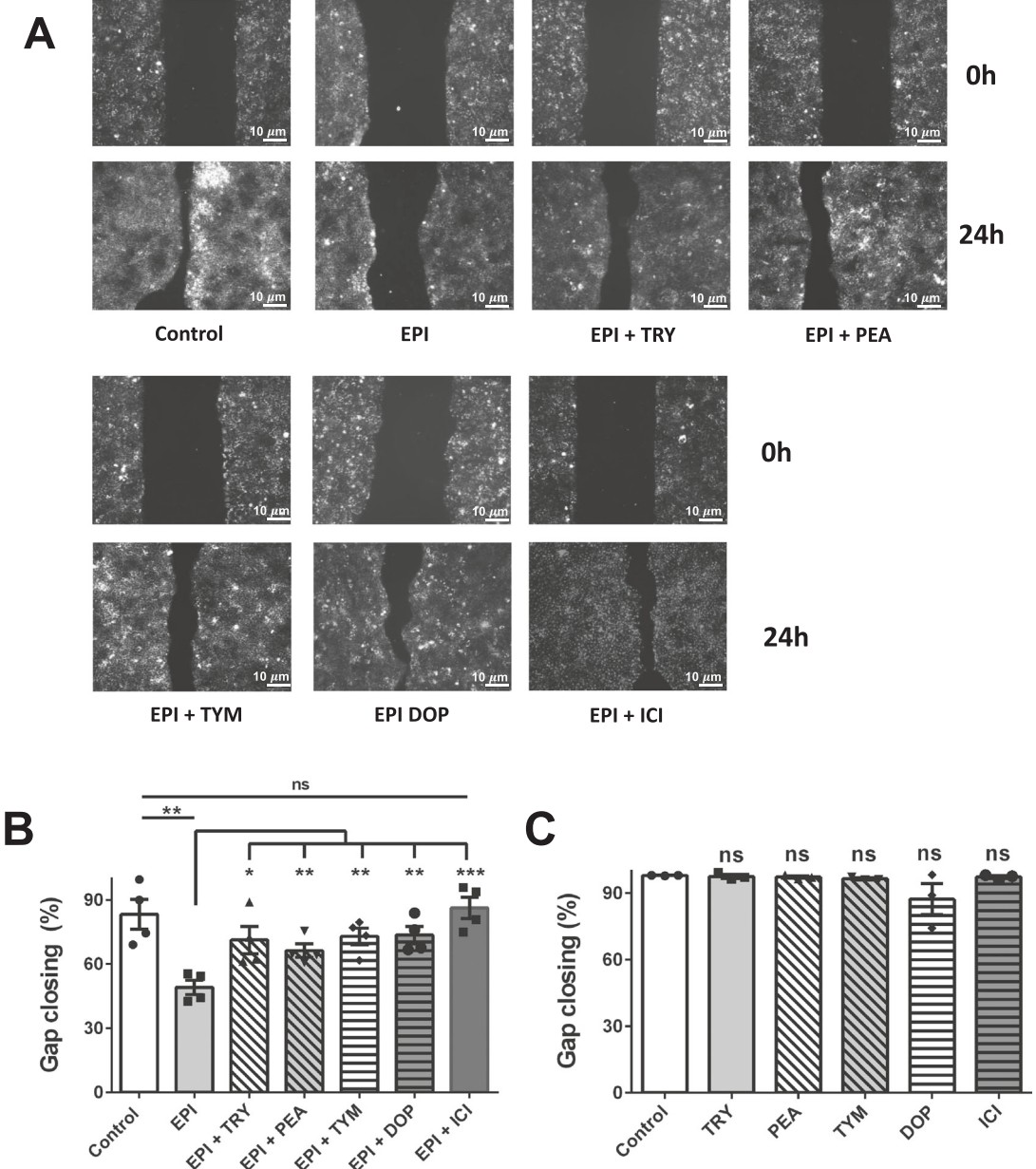

**Fig. 2 TA and DOP accelerate wound healing in vitro in a wound-mimicking assay. a** TA and DOP (each at 25 μg/ml) were added into the HaCaT cell culture seeded with a gap in the presence of EPI (25 μg/ml). ICI 118551 was used as positive control. Microscopic images of the gaps were taken at time 0 h and 24 h. **b** The gap closing, which was calculated using ImageJ[67], revealed that TA and DOP abrogated the migration-inhibiting effect of EPI. **c** In the absence of EPI, the TA, DOP and ICI showed similar gap closing as control. Each data point is the mean value ± SEM from 4 independent replicates for (**b**) and 3 independent replicates for (**c**), $*p < 0.05$; $**p < 0.01$; and $***p < 0.001$, data were analyzed using Student's $t$ test. Source data are provided as a Supplementary Data 1 file.

whether TA, DOP, and ICI can counteract this activity. Indeed, in the presence of EPI alone, the relative cAMP level was increased by approximately 50% compared to the control, while the addition of TA, DOP, and ICI caused a decrease ($p < 0.01$) in the cAMP level (Fig. 3b).

In contrast, the relative F-actin level, which positively correlates to the migration[36], was decreased by the addition of EPI and this effect was partially abrogated by TA and DOP (Fig. 3c). In the absence of EPI, TA, and DOP showed no effect on cAMP and F-actin level, while ICI decreased the cAMP level and increased the F-actin significantly ($p < 0.01$) compared to the control. This is due to the inverse agonist effect of ICI[31,32] (Supplementary Fig. 4a, b). As expected, in the presence of the

β-AR blocker, ICI, the effect of EPI was almost completely abrogated. Our data show that TA and DOP are only able to partially abrogate the EPI effect.

**TA and DOP affect neither effect cell proliferation nor intracellular Ca$^{++}$ level.** We performed HaCaT proliferation assay in the presence of EPI (25 μg/ml) alone and together with TA, DOP, and ICI (each at 25 μg/ml). EPI alone caused an increase in cell proliferation by about 50%. The addition of TA and DOP to the EPI treated cells had no effect (Fig. 4a). TA and DOP showed no difference to the control level in the absence of EPI (Fig. 4a, second half).

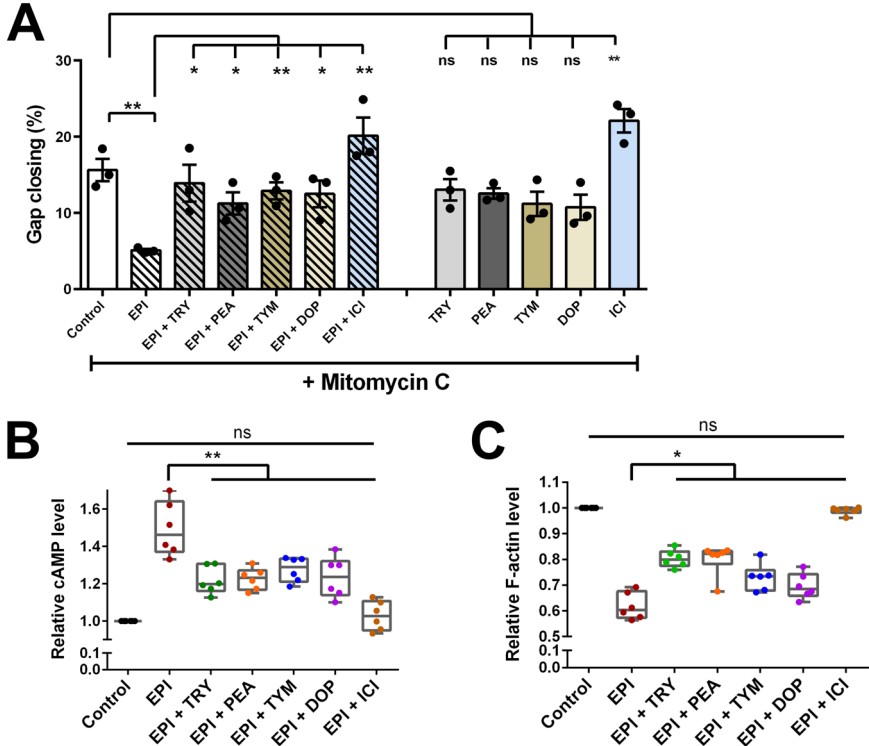

**Fig. 3 TA and DOP partially abrogate the effect of EPI in inhibiting HaCaT cells migration by affecting cAMP level and F-actin formation. a** HaCaT cells migration was examined using a similar method as the wound healing assay with the addition of Mitomycin C (10 μg/ml). TA, DOP and ICI (25 μg/ml) enhanced the HaCaT cells migration in the presence of EPI (25 μg/ml). TA and DOP alone did not show any significant effect while ICI boosted the HaCaT cells migration. **b** HaCaT cells was incubated with EPI (25 μg/ml) and TA, DOP and ICI as positive control (25 μg/ml) separately for 2 h. TA, DOP and ICI were added 30 min prior to the addition of EPI. EPI increased the intracellular cAMP level of HaCaT cells but the addition of TA, DOP and ICI, decreased the cAMP level compared to HaCaT cells treated with EPI alone. **c** EPI (25 μg/ml) decreased the F-actin level compared to the control. The addition of TA, DOP and ICI (25 μg/ml) in the presence of EPI, increased the F-actin level compared to the treatment with EPI only. For all graphs, each data point is the mean value ± SEM from 3 independent replicates for (**a**) and from 6 independent replicates for (**b**), *$p < 0.05$; **$p < 0.01$, data were analyzed using Student's $t$ test. Source data are provided as a Supplementary Data 1 file.

Considering that one of the factors that regulate the cell proliferation is intracellular $Ca^{++}$ level[37–39] we investigated the potential effect of TA and DOP on intracellular $Ca^{++}$ level. As shown in Fig. 4b, EPI increased the intracellular $Ca^{++}$ level after 3 min, whereas the addition of TA and DOP did not alter the $Ca^{++}$ level. Only the ß-blocker ICI decreased the $Ca^{++}$ level to the control value (Fig. 4b). The lack of responsiveness of TA and DOP to cell proliferation and the intracellular $Ca^{++}$ level in the presence of EPI confirms that TA and DOP do not affect cell proliferation. In the absence of EPI, the $Ca^{++}$ level was not affected by TA, DOP, and ICI (Supplementary Fig. 4c).

These results show that EPI impairs wound healing (gap closing) in keratinocytes and that TA and DOP abrogate the effect of EPI. Furthermore, the effect of the TA and DOP is not based on cell proliferation but rather on cell migration.

**TA and DOP produced by skin staphylococci accelerate the wound healing in a murine model.** Our next question was whether the positive effect of TA and DOP on wound healing can also be observed in vivo. It is well known that one of the stress responses to the wound is the systemic and local increase of EPI, which impairs wound healing[40,41]. Therefore, we investigated the effect of TA, DOP, and ICI on wound healing in a murine model. Cutaneous wound healing experiments were performed in DDY mice with two circular full-thickness wounds on the back of each mouse. Each mouse was treated both on one side with TA or DOP and the other side with water as control starting from day 0 post-wounding. ICI was used as a positive control. The wound

diameter was measured by calculating the mean of diameters measured from two sides of one wound (Supplementary Fig. 6). This procedure was chosen because the wound area was not always circular. The wounds treated with TA, DOP or ICI (25 μg/ml) closed faster than the controls (Fig. 5a and Supplementary Fig. 4). This confirms that the positive wound healing effect observed with keratinocytes also applies for the mouse model.

To mimic the natural condition where TA-producing staphylococci and AAA are present on the skin, we applied *S. epidermidis* O47 as a model for skin staphylococci on the wound murine model with the application of phenylalanine. It resulted that wounds with *S. epidermidis* O47 wild type closed faster than wounds with *S. epidermidis* O47 Δ*sadA* (Fig. 5b). This result suggests that the TA-producing skin staphylococci play a role in accelerating wound healing.

## Discussion
The fact that many staphylococcal species are able to produce TA with the help of the enzyme SadA is a new insight[8]. Originally, we investigated the occurrence of TA-producing staphylococci in the gut of human volunteers and found that the TA-producing strains were predominant among the isolated staphylococcal strains. This finding suggests that such strains most likely have an advantage, which to this time is unclear.

Here we addressed the question whether TA-producing staphylococci are prevalent on our skin and what effect TA production has for our skin. The skin is a classical habitat for members of the *Staphylococcacea* as already described by Kloos

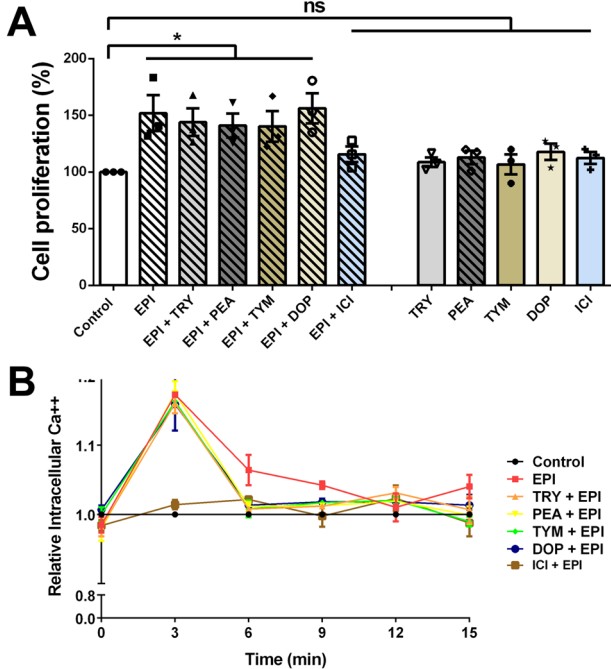

**Fig. 4 TA and DOP did not inhibit the HaCaT cell proliferation and the increase of intracellular Ca++. a** HaCaT cells proliferations were measured using MTT assay. The presence of EPI (25 μg/ml) increased the cell proliferations compared to control. The addition of TAs and DOP (25 μg/ml) showed no significant effect. **b** The intracellular Ca++ was measured using fluorescence-based assay. HaCaT cells was incubated in the presence of EPI (25 μg/ml) alone, and EPI with TA, DOP, and ICI. EPI increased the intracellular Ca++ level with and without TA and DOP at 3 min ICI nullified the effect of EPI both in terms of cells proliferation and increase in intracellular Ca++. For all graphs, each data point is the mean value ± SEM from three independent replicates, $*p < 0.05$; $**p < 0.01$; and $***p < 0.001$, data were analyzed using Student's $t$ test. Source data are provided as a Supplementary Data 1 file.

and Schleifer. The authors mapped and characterized the staphylococcal species on human skin[42,43], and presented a body atlas with the distribution of live staphylococcal species on various regions of the adult skin[44]. They already described that staphylococci, coryneforms, micrococci, and bacilli were the most predominant and persistent bacteria isolated from the head, legs, and arms. Later characterizations using genomic approaches revealed a much greater diversity of bacteria than that by the culture-based methods, but nevertheless, they come to a similar topographical distribution of bacteria on skin sites[45,46]. Due to the high abundance of staphylococci on the skin, we choose to take samples from the forearm (antecubital fossa) of 28 volunteers and plated them on a staphylococcal selective SK-salt agar. 20% of the 900 isolated staphylococcal colonies produced TA and the majority belonged to *S. epidermidis*, a species in which we found that >50% of the strains carried the *sadA* gene.

The key question in this study was whether TA can be produced by the skin bacteria and if so, the possible role they play on the skin. In staphylococci, we have found that TA are produced if the environment contains AAAs. There is no evidence that AAA are used for protein biosynthesis, instead they are imported, decarboxylated by SadA and excreted as TA[8]. Therefore, TA should be produced if sufficient AAAs are present on the skin. This is indeed the case as human sweat contains a comparatively high content (in the range of 300 μM or about 60 μg/ml) of AAAs[47–51]. In this context, approximately 10 μg/100 cm² of AAA and 5 μg/100 cm² of TA can be detected on our skin

(Supplementary Table 1). For this reason, it is expected that *sadA*-expressing staphylococci are able to produce TA on our skin. During the course of the investigation of the possible role TA–DOP–ICI might play on our skin, we found out that they accelerate wound healing.

After an injury, keratinocytes migrate over the wound bed to repair a wound. However, a wound triggers a stress reaction that leads to the production of various stress hormones such as EPI and cortisol which delay wound healing[28,29,41]. The enzymes necessary for both cortisol and EPI synthesis are located in the epidermis. Keratinocytes synthesize both hormones locally[41,52,53]. Keratinocytes also express beta-2-adrenergic-receptor (ß2-AR), a receptor for the stress hormone EPI, also known as adrenaline. Local cortisol and EPI synthesis impair keratinocyte migration and delay re-epithelialization[52,54]. Activation of β2-AR inhibits keratinocyte migration in a cAMP-independent manner[55], while antagonists of the ß2-AR accelerate wound healing[56]. The concentration of EPI that we used (25 μg/ml = 136 μM) is above the physiological concentrations found in plasma of mice with a wounded skin (18 ng/ml or 0.1μM)[41]. However, it could be that HaCaT cells express lower concentrations of ß2-AR. If this is the case, they would need a higher concentration of ligands to be responsive. On the other hand, we show in our mouse model (with the endogenously produced EPI) the effectiveness of TA.

The interaction of TA with the TA-associated receptors (TAARs), suggests that TA may also have functions that are entirely independent of the classical neurotransmitters[5,6]. Indeed, in co-stimulation studies have found that TYR and PEA are partial allosteric antagonists of ß1- and ß2-AR[11]. However, if TA are ß2-AR antagonists, then they should also accelerate wound healing.

To analyze the impact of TA–DOP–ICI on wound healing we have chosen two classical approaches. We carried out an in vitro migration or wound healing assay with HaCaT cells, a human keratinocyte cell line that expresses β2-AR[27]. In the presence of EPI, TA–DOP–ICI accelerated gap closing ($p < 0.05$) (Fig. 2a, b). No effect of TA–DOP–ICI was observed in the absence of EPI (Fig. 2c).

It is important to discriminate between the effect of proliferation and migration of adjacent cells on wound closure. Therefore, we investigated the wound healing of HaCaT cells in the presence of mitomycin C, a potent DNA crosslinker and inhibitor of cell proliferation. Comparison of Fig. 2 and Fig. 3 shows that in the presence of mitomycin C the gap closing was generally decreased from 85% to about 15%. However, independent of the presence or absence of EPI, TA–DOP–ICI always increased gap closing to almost the control level (Fig. 3a).

These result show two things: firstly, EPI worsens wound healing in the presence of mitomycin C. In addition to the inhibition of cell proliferation by mitomycin, cell migration is now inhibited by EPI. Similar effects were reported for catecholamines (isoproterenol, EPI, and norepinephrine), which delay wound closure by inhibiting epidermal cell migration[57]. Secondly, TA–DOP–ICI counteracted the negative effect of EPI on gap closing, indicating that they activate cell migration.

To further confirm that TA–DOP–ICI abrogate the negative effect of EPI on cell migration, we investigated the cAMP and F-actin levels in cytoplasm. It has been the accepted dogma since the 1960s that β2-receptor activation causes an increased intracellular cAMP levels due to the stimulation of adenylate cyclase[58,59]. As expected, EPI increased the plasma cAMP level in HaCaT cells, and this increase was counteracted by TA–DOP–ICI, indicating that they antagonize the EPI effect (Fig. 3b). Similar effect was observed with F-actin (Fig. 3c). EPI decreased the F-actin level, while TA–DOP–ICI relieved this blockade. The EPI-induced decrease of the F-actin level is responsible for the inhibition of cell migration.

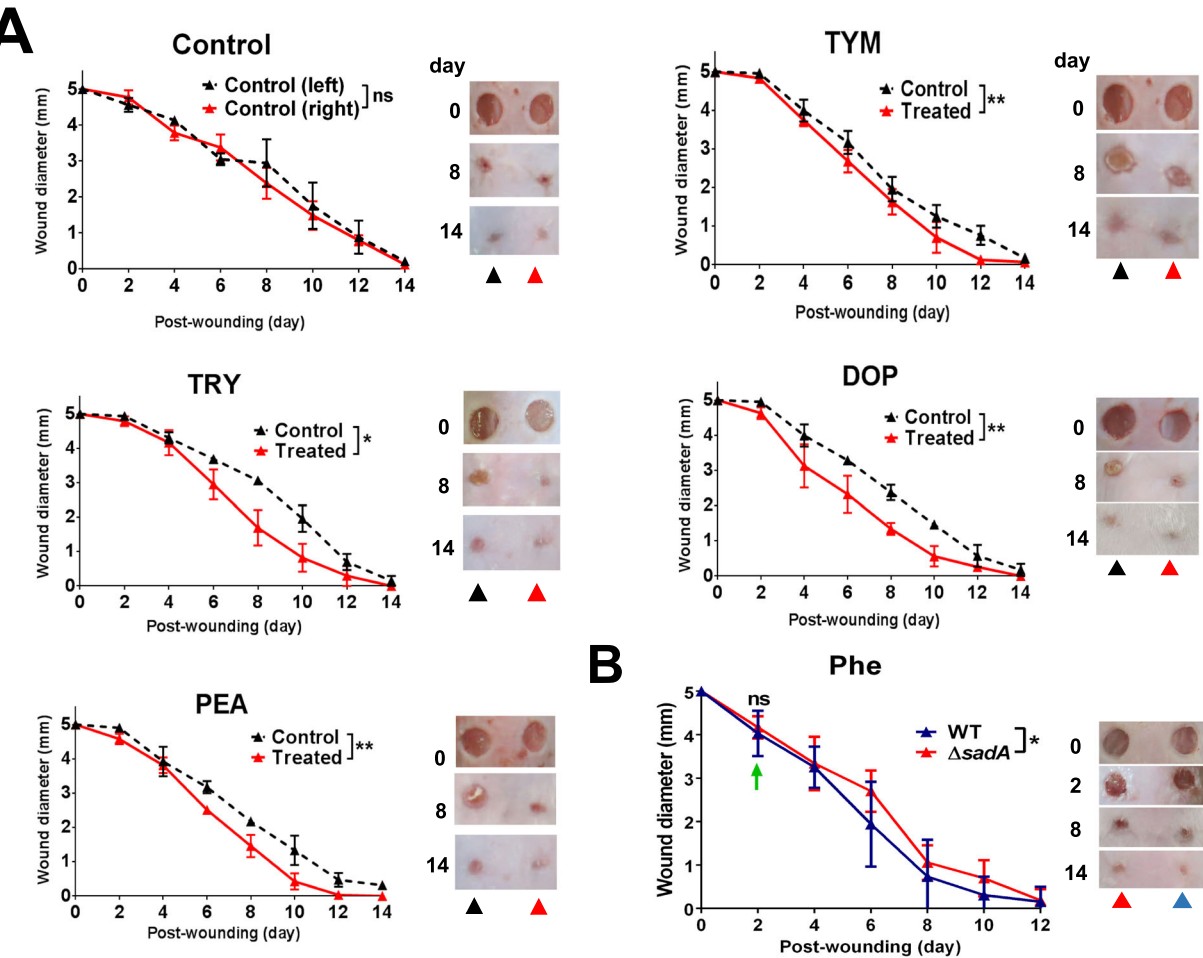

**Fig. 5 Effect of TA on wound healing in mice.** Mice were shaved and wounded on their back with 2 biopsy punches for each mouse. **a** TRY, PEA, TYM, and DOP (each 25 μg/ml) with total volume of 10 μl were applied topically daily starting from day 0. The diameter of the wounds was measured every 2 days. The wounds treated with TRY, PEA, TYM, and DOP showed a faster wound closing than the untreated wounds (control). **b** *S. epidermidis* O47 WT and Δ*sadA* were applied on and around the wound at cell density 5000 CFU/100 cm$^2$ on 2 days post-wounding (green arrow). In addition, 10 μl phenylalanine (Phe) 50 μg/ml was also applied daily. The diameter of the wounds on 2 days post-wounding before application of *S. epidermidis* O47 were not significantly different. Wounds treated with *S. epidermidis* O47 WT showed faster wound closure than those treated with *S. epidermidis* O47 Δ*sadA*. For all graphs, each data point is the mean value ± SEM from three independent replicates for (**a**) and six independent replicates for (**b**), *$p < 0.05$ and **$p < 0.01$, data were analyzed using paired Student's *t* test. Source data are provided as a Supplementary Data 1 file.

It has been reported that EPI stimulates cell proliferation[60] and increases the intracellular calcium level[61]. Therefore, we tested TA–DOP–ICI whether they negate EPI's activity on cell proliferation and intracellular calcium level (Fig. 4a, b). However, there was no such activity observed, indicating that TA–DOP are not involved in cell proliferation. In our studies, ICI was also included as a control since it is a selective ß2-AR antagonist or beta blocker[31]. Its effect to counteract EPI-induced activation of ß2-AR was always stronger than that of TA–DOP, suggesting that ICI is a full ß2-AR antagonist, while TA–DOP are partial antagonists.

As all these studies were done in vitro with keratinocytes we examines the activity of TA–DOP in vivo using a mouse model. In agreement with in vitro studies, the wound healing was accelerated by TRY, DOP, PEA, TYM, and ICI ($p < 0.05$) (Fig. 5a and Supplementary Fig. 5).

Cogen et al. [62] stated that skin microbiota are usually regarded as pathogens, potential pathogens or innocuous symbiotic organisms. Indeed, it has long been assumed that a normal skin microbiota has rather a protective rather than destructive function on our skin. For example, a commensal strain of *S. epidermidis*

protects again skin neoplasia[63]. Here we show that the TA produced by skin bacteria accelerate wound healing (Fig. 5b). A wound represents a stress situation which is accompanied by the production of various stress hormones such as EPI and cortisol which delay wound healing by inhibiting cell migration due to activation of ß2-AR. Members of *Staphylococcaceae*, particularly the classical skin bacterium *S. epidermidis*, produces TA, which act as neurotransmitter or neuromodulators. While studying the function of TA on our skin, we found that they abrogate the negative effect of EPI on wound healing thus accelerate wound healing. Surprisingly, when we applied the *S. epidermidis* strains 2 days post-wounding at and around the wound area, 95% of the mice showed no infection symptoms such as inflammation or necrosis, suggesting that commensal *S. epidermidis* strains evolved strategies to co-exist rather than being a symbiote on our skin.

## Methods
**Skin swab samples collection, *sadA*-expressing staphylococci screening and AAA and TA quantification of skin swab samples**. We collected the skin swab samples from 28 healthy and random subjects by swabbing the antecubital fossa part using sterile swab sample collector and sterile phosphate buffer saline (PBS).

Collected skin swab samples were then spread on SK salt agar medium[8] and incubated at 37 °C for 48 h. The colonies were then picked, inoculated in basic medium (BM) in 96-well plates and incubated at 37 °C for 24 h with shaking at 150 rpm. The collected supernatants were analyzed by reversed-phase HPLC (RP-HPLC) at room temperature with an Eclipse XDB-C18 column (4.6 × 150 mm; 5 μm) (Agilent) installed together with an analytical guard column for Eclipe XDB-C-18 (4.6 × 12.5 mm; 5 μm) (Agilent), with a 15-min linear gradient of 0.1% phosphoric acid to acetonitrile and 5 min post time washing with 0.1% phosphoric acid. The flow rate used was 1.5 mL/min; the sample volume that was injected was 10 μl. We used diode array detector (DAD) at 210 nm and 360 nm as reference. The simplified workflow is illustrated in Fig. S1. While for AAA and TA quanti-fication on skin swab samples, we resuspended the samples in sterile PBS and analyzed using RP-HPLC with the same method as for screening.

**Skin staphylococci identification**. We identified the species of the isolated TA producing staphylococci by growing the strains on the selective agar separately and incubated overnight at 37 °C. We then picked the colonies, resuspended in PBS and isolated the genomic DNA using Quick-DNA Microprep Kit (Zymo Research). We then amplified the 16s rRNA gene using specific primers (Supplementary Tabel 1) and Q5 polymerase (New England Biolabs). We purified the PCR product using Illustra GFX DNA and Gel Band Purification Kit (GE healthcare) and sequenced it (GATC). We identified the species by performing BLASTN from the obtained sequence of amplified 16s rRNA gene against the 16s rRNA database in NCBI.

**Construction of *sadA* deletion mutant in *S. epidermidis* O47**. Construction of *S. epidermidis* O47 Δ*sadA* was carried out using plasmid pBASE6[64]. The 1 kb upstream downstream region of *sadA* were amplified using appropriate primers (Supplementary Table 2) and ligated with linearized pBASE6 (*Eco*RV restriction site) using Hi-Fi DNA Assembly Master Mix (New England Biolabs) and trans-formed into *E. coli* C2987 (New England Biolabs) and then into *E. coli* DC10B chemically competent cells. Colonies with the desired plasmid were picked and the respective plasmid was transformed into *S. epidermidis* O47 by protoplast trans-formation. Mutagenesis was conducted as reported by[65]. Mutants were confirmed using PCR for the gene deletion and HPLC analyses for the overnight supernatants.

**Bioinformatic analysis**. Homologs of *SadA* from *S. pseudintermedius* ED99 were identified within the species *S. epidermidis*. To find homologs on the strain level tblastn of NCBI's Microbial BLAST was used and all complete genomes of *S. epidermidis* were used as a database. As a query the *SadA* homolog of the *S. epidermidis* strain ATCC 12228 (NC_004461.1, genomic position 107418-108839) was used as a query sequence.

**In vitro wound healing assay with HaCaT cells**. The in vitro wound healing assay with HaCaT cells was carried out essentially as described by Jonkman et al.[66] HaCaT cells were seeded in culture-insert 2 well in μ-dish$_{35 mm,low}$ (Ibidi) with a cell density 6 × 10$^5$ cells/ml and in 70 μl volume in DMEM medium with 10% fetal bovine serum (FBS) and antibiotic mix and incubated at 37 °C in 5% CO$_2$ for 24 h prior to the assay. After 24 h incubation, the culture-inserts were gently removed, and the cell culture was washed with PBS twice and 1 ml of DMEM medium was added into the μ-dish. Gap closing was microscopically monitored (Leica DMLB) with ×100 magnification (×10 objective lens and ×10 ocular lens magnification). The TA, DOP and ICI were added at a concentration of 25 μg/ml each and incubated for 30 min prior to the addition of EPI (25 μg/ml). Cell cultures were then incubated for 24 h and microscopic images were taken. The gaps were ana-lyzed using ImageJ as described in ref. [67] and wound closing was calculated by subtracting the final gap area from initial gap area. Cell migration assays were carried out in a similar way as described for the in vitro wound healing assay, but 10 μg/ml of Mitomycin C was added as a cell proliferation inhibitor. We performed at least three times independent replications for each experiments.

**Cell proliferation assay**. Prior to the cell proliferation assay, HaCaT cells were seeded in a 96-well microtiter flat-bottom plate with 5 × 10$^4$ cells/well and incu-bated 24 h at 37 °C in 5% CO$_2$. The HaCaT cells were treated with the neuro-chemicals (TRY, PEA, TYM, and DOP) and ICI 118155 at a final concentration of 25 μg/ml, with and without EPI (25 μg/ml). The cell proliferation assay was per-formed using the Cell Proliferation Kit I (MTT; Roche, Germany) according to the protocol provided by the company. We performed at least 3 times independent replications for each experiment.

**cAMP level measurement of HaCaT cells**. Prior the cAMP measurement, HaCaT cells were seeded in black 96-well plate flat bottom with 1 × 10$^5$ cells per well and incubated overnight in DMEM medium with 10% FBS and antibiotic mix and incubated at 37 °C in 5% CO$_2$. Cells were then treated with TA, DOP and ICI (25 μg/ml) for 30 min prior to the addition of EPI (25 μg/ml) and incubated further for 1.5 h. cAMP levels were measured using cAMP Glo$^{TM}$ Assay (Promega) according to the protocol provided by the company. We performed at least 3 times independent replications for each experiment.

**F-actin level**. Prior the cAMP measurement, HaCaT cells were seeded in black 96-well plate flat bottom with 1 × 10$^5$ cells per well and incubated overnight in DMEM medium with 10% FBS and antibiotic mix and incubated at 37 °C in 5% CO$_2$. Cells were then treated with TA, DOP and ICI (25 μg/ml) for 30 min prior to the addition of EPI (25 μg/ml) and incubated further for 1.5 h. The treated cells were then washed with DPBS, permeabilized with 0.1% (v/v) Triton X-100, washed again with DPBS, stained with ActinGreen™ 488 ReadyProbes® (Thermo Fischer) for 30 min and washed again 3–5 times with DPBS with soaking duration of 5 min for each washing step. The relative fluorescence intensity was measured at 495 nm for the excitation and 518 nm for the emission using Tecan Infinite M200. We performed at least 3 times independent replications for each experiment.

**Intracellular calcium assay**. We seeded the HaCaT cells in 96-well microtiter flat bottom plate with 5 × 10$^4$ cells/well and incubated for 24 h at 37 °C in 5% CO$_2$. The intracellular calcium measurement assays were performed using Fluo-8 calcium flux assay kit—no wash (Abcam) according to the protocol provided by the company. The HaCaT cells were treated with the neurochemicals (TRY, PEA, TYM, and DOP) and ICI, with and without EPI. We performed at least 3 times independent replications for each experiment.

**In vivo murine model**. For wound healing experiments we followed essentially the protocol for cutaneous wound healing in a murine model as described Ganuli-Indra[68]. Six to eight weeks old male of DDY mice were used in wound healing experiments. Prior to the wounding, mice were anesthetized with ketamine/xyla-zine (10:1) with a dose of 0.04 mg/g mouse weight. The back of the mice was shaved and two circular full-thickness wounds (5 mm in diameter) were made on the back skin of each mouse using a skin biopsy punch. Two wounds were made on each mouse (right and left) so that the control and treatment were on the same mouse. The neurochemicals TRY, PEA, TYM, and DOP as well as ICI 118551 at a concentration of 25 μg/ml, respectively, and a total volume of 10 μl each was applied topically on one wound (right), water was applied as control on the other wound (left). The application of the neurochemicals was done daily, and the wound diameters were measured every two days until 14 days post-wounding. We used three mice for each variation in these experiments. For wound healing experiments using bacteria, we used *S. epidermidis* O47 WT and its Δ*sadA* deletion mutant as model bacteria. At two days post-wounding, we applied *S. epidermidis* O47 WT (on the right wound) and Δ*sadA* (on the left wound) with a cell density of 5000 CFU/100 cm$^2$ at the wound area. At the same time, we applied phenylalanine (50 μg/ml) with a total volume of 10 μl on wound daily. The wound diameters were measured every two days until 14 days post-wounding. We used six mice in this experiment.

**Statistics**. Normal distributions were analyzed by Student's *t* test. Statistical ana-lyses were performed with GraphPad Prism software, with significance defined as $p < 0.05$; n represents independent biological replicates.

**Ethical statement**. The collection of human skin swab samples was approved by the Ethic Commission of the University of Tübingen (Approval no. 320/2017BO2). Skin swab samples were obtained from 28 adult volunteers. All samples were anonymized and obtained with written consent from the volunteers. The wound healing experiment using mice were approved by the Ethic Commission of Faculty of Veterinary Medicine, Universitas Airlangga, Surabaya, Indonesia (Approval no. 2.KE.172.09.2019).

**Reporting summary**. Further information on research design is available in the Nature Research Reporting Summary linked to this article.

## Data availability

All relevant data are available from the authors upon request. The 16s rDNA sequences are uploaded to NCBI with accession number MT445230-MT445414. Source data underlying plots shown in figures are provided in Supplementary Data 1.

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

## Acknowledgements

The authors thank Dr. Sook-Ha Fan for proofreading the manuscript and Dr. Surasak Jittavisutthikul and Mulugeta Nega for the valuable suggestions. This work was supported by the Deutsche Forschungsgemeinschaft (DFG) SFB766 and SFB/TRR24 to F.G., and by infrastructural funding from the Deutsche Forschungsgemeinschaft (DFG), Cluster of Excellence EXC 2124 Controlling Microbes to Fight Infections. It was also supported by the Ministry for Science, Research and the Arts of Baden-Wuerttemberg (MWK) "AntibioPPAP". PMT held a postdoctoral Alexander von Humboldt fellowship.

## Author contributions

F.G., D.H. and A.L. designed the study. A.L., P.E., M.M., P.M.T., K.N., D.H. and F.G. designed the experiments. A.L. performed all experiments except the in vivo wound healing which was carried out by M.Z.M. and S.Y., the bioinformatic analysis was performed by S.Z. and F.G., A.L. and S.Z. wrote the manuscript.

## Competing interests

The authors declare no competing interest.
