## [Peer Review File · Communications Biology]

Reviewers' comments:

Reviewer #1 (Remarks to the Author):

The paper by Luqman et al. describes a hitherto unknown role of skin commensals *S. epidermidis* in skin healing after injury. The observations are based on the assumption of a complex system of local molecular interactions on the epidermic layer of the skin. Upon the observation that certain *S. epidermidis* isolates contain SadA, a decarboxylase conferring the decarboxylation of aromatic amino acids into trace amines TAs and of L-DOPA into dopamine, they assumed a role of TAs on keratinocytes. The mode of interference was inferred from previous observations, i.e. the well-known role of beta-agonists (epinephrine) on the beta2-adrenergic receptor with the consequence of intracellular cAMP increase and F-actin decrease resulting in reduced cell migration. In essence, they hypothesized that the addition of TAs or DOP might antagonize the inhibitory EPI effect on keratinocyte migration and wound closure.

To this end, they employed in vitro gap closure experiments, determinations of intracellular cAMP and F-actin levels, cellular proliferation, and finally an in vivo model of mice wound healing employing either TA's and DOP, or colonizing mice with *S. epidermidis* SadA WT and a Δ SadA mutant, respectively. From their data, they clearly conclude that skin-locally produced TA's might well abrogate stress (adrenalin)-induced inhibition of wound healing, and that *S. epidermidis* isolates producing such TA's from abundant aromatic AAs may therefore have a protective effect on skin wound healing. This study has therefore a large potential for understanding the role of certain skin commensals, of the skin microbiota on the integrity of the human skin, and the potential for resulting translational applications.

Overall, the study is very carefully designed and comprehensive. After demonstrating the prevalence of SadA producing skin isolates on the skin of volunteers, authors performed the aforementioned experiments on HaCaT keratinocyte gap closure with the appropriate agonists, antagonists, and notably, appropriate controls. The results show not only statistically highly significant, but likely biologically relevant abrogation of gap closure as a result of cell migration (not of keratinocyte proliferation!), and indicate that the restoration of intracellular F-actin levels upon EPI inhibition might be the cause. These results as well as the animal experiments are clear, well-described, and conclusions are supported by data. Overall, in addition to a clear conception of the study, these meaningful in vitro and in vivo data make the study highly valuable.

I have only minor comments:

- In some portions of the manuscript, the use of English language should be carefully checked and corrected. For instance:

- lines 59-61: contains no verb

- line 61: "comparing"... "showed"

- line 96: comma after "skin colonizers"

- line 100: comma after "microflora"

- line 290/291: "complex processes occur" .. "are still not... understood"

- line 291: Stand-alone sentence without verb

- line 293: "situating"

- The agonist/antagonist role of TAs with different receptor types should be emphasized and explained. In the abstract, the reader is taught about the ANTAGONISTIC effect of TA on β 2-AR, then (line 106) on the AGONISTIC effect on α 2-AR, then again (line 149) on the β 2 ANTAGONISM while subsequently (lines 156ff.) hypothesizing on an EPI abrogating effect of TAs being α 2 AGONISTS. An introduction into α 2/ β 2 adrenergic receptor differences with respect to EPI effect and TA/DOP abrogation would be extremely useful to prevent confusion for the reader

- The discussion can be largely shortened. Lines 262-306 either repeat or reemphasize issues which should be presented in the introduction.

- On the other hand, the reader awaits an answer to the overall question (line 94) which advantage bacteria get by the synthesis of TA. What would be the selection advantage for skin commensals such as staphylococci when injured skin is more readily repaired? Do staphylococci preferentially/exclusively reside on intact skin? This study is not expected to give definitive answers, but as the question has been posed, at least some critical considerations might be made

in the discussion.

Reviewer #2 (Remarks to the Author):

The manuscript by Luqman et al explores a new paradigm for regulation of aspects of wound healing by trace amines, such as those generated by certain bacterial strains. This is a novel concept and findings may contribute to advances in understanding the interplay between commensals and wound healing. However, at the current stage, there are a number of major concerns that preclude valid interpretations of the presented work. Minor concerns are also noted.

Major concerns

- 1- Conclusions from HPLC analysis are impossible without full description of methods. Methods should be given in enough detail so that the procedures can be replicated by other labs. Besides the parameters given, the investigators should also mention the column temperature setting (if thermostatted), injection details (injection volume, full-loop or partial-loop injections), the detector and settings used, and other important details. Sample work-up should also be described.
- 2- Conclusions from the animal wound studies are not supported using the method the investigators chose (measuring two diameters across the wound). The authors state that this method is more reliable than imaging the wound surface, but have no support for this statement. Indeed, the images of the wounds presented show irregular wound shapes, so measuring at different parts of the wound would yield different numbers. In fact, histological evaluation of the epidermal wound gap is considered to be the most reliable way to determine wound size.
- 3- Importantly, one cannot ascribe the effects of TA to blockade of the beta-adrenergic receptor. TA's may be functioning through their cognate receptors, and thus the presence / absence of specific TA receptors on skin cells should be examined. Additionally, TA's can interact with alpha adrenergic receptors so the effects noted could be ascribed their activation. To discriminate between these possibilities, antagonists of the alpha receptor subtypes should be included. Because some unexpected ICI results are attributed to it being an "inverse agonist" other beta antagonists devoid of inverse agonist activity should be tested.
- 4- Also importantly, since staph aureus is a known pathogen and inhibits healing, the ability of those species to generate TAs should be reported here.
- 5- The concentration of epinephrine used (136 μ M) is far above physiologic or even stress levels (10-50 nM) and far above the concentrations that other investigators of effects of epinephrine on skin cells have reported (10nM- 1 μ M). With these high levels of epinephrine, it is unclear if the observations are related to off-target effects. These experiments should be repeated with more relevant concentrations.
- 6- Line 480 "The collected skin swab samples were analyzed using reverse phase HPLC..." is unclear. Were TA measured directly from the swabs, or does this refer to supernatants from the swab inoculated cultures?
- 7- Calcium concentrations were measured only at one time point. It is not clear if there is an early spike then then return to baseline over time, as with other agents that induce increases in keratinocyte Calcium levels.

Minor concerns:

- 1- The use of mitomycin C in scratch wound assays is routine and does not need an entire paragraph to discuss the rationale.
- 2- Please cite more recent maps of skin microbiome (eg Grice et al) rather than work from 1975.
- 3- It is unclear if all 900 isolates were tested by both HPLC and PCR.
- 4- Many of the findings have already been published by others (decrease in keratinocyte migration

by epinephrine/catecholamines; increase in skin wound healing by ICI, figure S4) and thus should not be included here. Reaffirmation of prior investigators' findings should be more clearly pointed out.

5- Line 330: error in calling cell cytoplasm "plasma"

6- DDY mice. DDY not identified. Sex not included.

7- Water as a control, is hypotonic and may be damaging the wound bed contributing to decreased healing seen in the controls relative to TA treated. Saline is better control.

8- Needs native English speaker editing. The manuscript is wordy and repetitive. Needs major shortening.

Reviewers' comments:

Reviewer #1 (Remarks to the Author):

The paper by Luqman et al. describes a hitherto unknown role of skin commensals *S. epidermidis* in skin healing after injury. The observations are based on the assumption of a complex system of local molecular interactions on the epidermic layer of the skin. Upon the observation that certain *S. epidermidis* isolates contain SadA, a decarboxylase conferring the decarboxylation of aromatic amino acids into trace amines TAs and of L-DOPA into dopamine, they assumed a role of TAs on keratinocytes. The mode of interference was inferred from previous observations, i.e. the well-known role of beta-agonists (epinephrine) on the beta2-adrenergic receptor with the consequence of intracellular cAMP increase and F-actin decrease resulting in reduced cell migration. In essence, they hypothesized that the addition of TAs or DOP might antagonize the inhibitory EPI effect on keratinocyte migration and wound closure.

To this end, they employed in vitro gap closure experiments, determinations of intracellular cAMP and F-actin levels, cellular proliferation, and finally an in vivo model of mice wound healing employing either TA's and DOP, or colonizing mice with *S. epidermidis* SadA WT and a Δ SadA mutant, respectively. From their data, they clearly conclude that skin-locally produced TA's might well abrogate stress (adrenalin)-induced inhibition of wound healing, and that *S. epidermidis* isolates producing such TA's from abundant aromatic AAs may therefore have a protective effect on skin wound healing. This study has therefore a large potential for understanding the role of certain skin commensals, of the skin microbiota on the integrity of the human skin, and the potential for resulting translational applications. Overall, the study is very carefully designed and comprehensive. After demonstrating the prevalence of SadA producing skin isolates on the skin of volunteers, authors performed the aforementioned experiments on HaCaT keratinocyte gap closure with the appropriate agonists, antagonists, and notably, appropriate controls. The results show not only statistically highly significant, but likely biologically relevant abrogation of gap closure as a result of cell migration (not of keratinocyte proliferation!), and indicate that the restoration of intracellular F-actin levels upon EPI inhibition might be the cause. These results as well as the animal experiments are clear, well-described, and conclusions are supported by data. Overall, in addition to a clear conception of the study, these meaningful in vitro and in vivo data make the study highly valuable. I have only minor comments:

- In some portions of the manuscript, the use of English language should be carefully checked and corrected. For instance:
- lines 59-61: contains no verb
- line 61: "comparing" ... "showed"
- line 96: comma after "skin colonizers"
- line 100: comma after "microflora"
- line 290/291: "complex processes occur" .. "are still not... understood"
- line 291: Stand-alone sentence without verb
- line 293: "situating"

We thank the reviewer for the comments. We have accepted all suggestions (highlighted in yellow).

- The agonist/antagonist role of TAs with different receptor types should be emphasized and explained. In the abstract, the reader is taught about the ANTagonistic effect of TA on $\beta\beta$ -AR, then (line 106) on the AGONISTIC effect on $\alpha\alpha$ -AR, then again (line 149) on the $\beta\beta$ ANTagonism while subsequently (lines 156ff.) hypothesizing on an EPI abrogating effect of TAs being $\alpha\alpha$ AGONISTS. An introduction into $\alpha\alpha/\beta\beta$ adrenergic receptor differences with respect to EPI effect and TA/DOP abrogation would be extremely useful to prevent confusion for the reader.

We added few sentences regarding this issue in the introduction part line 81-83.

- The discussion can be largely shortened. Lines 262-306 either repeat or reemphasize issues which should be presented in the introduction.

We deleted the passage on the advantage of internalization in colonization.

- On the other hand, the reader awaits an answer to the overall question (line 94) which advantage bacteria get by the synthesis of TA. What would be the selection advantage for skin commensals such as staphylococci when injured skin is more readily repaired? Do staphylococci preferentially/exclusively reside on intact skin? This study is not expected to give definitive answers, but as the question has been posed, at least some critical considerations might be made in the discussion.

We wrote a possible answer for that question in line 102-106. It is written as One reason for the predominance of TA-producing staphylococci in the gut could be that TA increase internalization of the bacteria into host cells 8. It is well documented that an increased internalization of the epithelial cells protects the bacteria from both the host immune system 16-18 and antibiotics 19,20.

Reviewer #2 (Remarks to the Author):

The manuscript by Luqman et al explores a new paradigm for regulation of aspects of wound healing by trace amines, such as those generated by certain bacterial strains. This is a novel concept and findings may contribute to advances in understanding the interplay between commensals and wound healing. However, at the current stage, there are a number of major concerns that preclude valid interpretations of the presented work. Minor concerns are also noted.

Major concerns

1- Conclusions from HPLC analysis are impossible without full description of methods. Methods should be given in enough detail so that the procedures can be replicated by other labs. Besides the parameters given, the investigators should also mention the column temperature setting (if thermostatted), injection details (injection volume, full-loop or partial-loop injections), the detector and settings used, and other important details. Sample work-up should also be described.

We thank the reviewer for the comment. We added more detailed method in the Material and Method part (line 386-391).

2- Conclusions from the animal wound studies are not supported using the method the investigators chose (measuring two diameters across the wound). The authors state that this method is more reliable than imaging the wound surface, but have no support for this statement. Indeed, the images of the wounds presented show irregular wound shapes, so measuring at different parts of the wound would yield different numbers. In fact, histological evaluation of the epidermal wound gap is considered to be the most reliable way to determine wound size.

We thank the reviewer for the comment. The problem was that the wound area was in many cases not completely circular and taking the photos at the same position every two days was complicated. Therefore, we have not chosen the area, which might have been the most easy way, but measured the diameter. So, we corrected the sentence: We have chosen this procedure because the wound area was not always circular (line 245).

3- Importantly, one cannot ascribe the effects of TA to blockade of the beta-adrenergic receptor. TA's may be functioning through their cognate receptors, and thus the presence / absence of specific TA receptors on skin cells should be examined. Additionally, TA's can interact with alpha adrenergic receptors so the effects noted could be ascribed their activation. To discriminate between these possibilities, antagonists of the alpha receptor subtypes should be included. Because some unexpected ICI results are attributed to it being an "inverse agonist" other beta antagonists devoid of inverse agonist activity should be tested.

HaCaT cells were reported to express beta-adrenergic receptor instead of alpha-adrenergic receptor. If there was interaction of TA with either alpha2-adrenergic receptor and TAARs, which all of them are GPCR, we should have observed a significant change in cAMP level in HaCaT cells. Our experiments, treating HaCaT with TA, showed no significant change in cAMP level (Figure S4 A). It suggests that there is no interaction between TA and either alpha2-adrenergic receptor or TAARs. Nevertheless, we tested pentholamine (PTL, an alpha antagonist) in *in vitro* wound healing experiments. The addition of PTL together with EPI showed no difference in comparison with EPI alone, suggesting that alpha2 adrenergic receptor does not play a significant role in wound healing. We also tested alprenolol (ALP, a neutral, non-selective beta antagonist used as a therapeutic) in *in vitro* wound healing experiments; ALP showed similar results as ICI in gap closing (new Figure S3B).

4- Also importantly, since staph aureus is a known pathogen and inhibits healing, the ability of those species to generate TAs should be reported here.

We found *S. aureus* in the skin swab samples, however, none of the isolates produced TAs (see Fig. 1C).

5- The concentration of epinephrine used (136 uM) is far above physiologic or even

stress levels (10-50 nM) and far above the concentrations that other investigators of effects of epinephrine on skin cells have reported (10nM- 1 uM). With these high levels of epinephrine, it is unclear if the observations are related to off-target effects. These experiments should be repeated with more relevant concentrations.

The concentration of EPI that we used (25 µg/ml = 136 µM) is above the physiological concentrations found in mice 18 ng/ml (0.1µM) plasma of wounded skin {Sivamani, 2009 #26}. However, it could be that HaCaT cells express lower concentrations of β2Ars. If this is the case they would need a higher concentration of ligands to be responsive. On the other hand we show in our mouse model (with the endogenously produced EPI) the effectiveness of TAs. Line 303-308

6- Line 480 “The collected skin swab samples were analyzed using reverse phase HPLC...” is unclear. Were TA measured directly from the swabs, or does this refer to supernatants from the swab inoculated cultures?

We have corrected the sentence to be more clear. It is now written as: “The collected the skin swab samples were resuspended in sterile PBS and analysed using reversed-phase HPLC.....”. The TA and AAA were measured from the resuspended skin swab.

7- Calcium concentrations were measured only at one time point. It is not clear if there is an early spike then then return to baseline over time, as with other agents that induce increases in keratinocyte Calcium levels.

We are very grateful for the reviewer’s suggestion. We made a new graph for intracellular Ca conc as shown in Fig 4B and Fig S4C.

Minor concerns:

1- The use of mitomycin C in scratch wound assays is routine and does not need an entire paragraph to discuss the rationale.

We thank the reviewer for the comment. We wrote short explanation about mitomycin C because it is important introduction for our results about the effect of TA in cells migration and proliferation separately.

2- Please cite more recent maps of skin microbiome (eg Grice et al) I rather than work from 1975.

We have cited 2 papers from Grice et al in the reference (reference number 46 and 47).

3- It is unclear if all 900 isolates were tested by both HPLC and PCR.

We mentioned in the results part that 900 isolates have been analysed by HPLC and 185 positive TA-producing isolates were identified using 16s (line 137-140).

4- Many of the findings have already been published by others (decrease in keratinocyte migration by epinephrine/catecholamines; increase in skin wound healing by ICI, figure S4) and thus should not be included here. Reaffirmation of prior investigators' findings should be more clearly pointed out.

We used EPI and ICI as controls for the experiment, so we can compare the effects of TA with the controls.

5- Line 330: error in calling cell cytoplasm "plasma"

We have corrected accordingly.

6- DDY mice. DDY not identified. Sex not included.

Due to the sex dimorphism in wound healing and we did not use sex as one of the variable in our experiments, we only used male DDY mice for our *in vivo* wound healing experiments.

7- Water as a control, is hypotonic and may be damaging the wound bed contributing to decreased healing seen in the controls relative to TA treated. Saline is better control.

We thank the reviewer for the comment. We used water as control and solvent TA in all *in vitro* experiments. So, in order to have similar condition as in our *in vitro* experiments, we also used water as control and solvent in our *in vivo* experiments.

8- Needs native English speaker editing. The manuscript is wordy and repetitive. Needs major shortening.

The ms was checked by the native English speaking co-worker, Dr. Sook-Ha Fan.

REVIEWERS' COMMENTS:

Reviewer #1 (Remarks to the Author):

All my comments are met

Reviewer #2 (Remarks to the Author):

Many of the comments of this reviewer have been addressed.

However the major concern remains: that is the over-interpretation of wound diameter as an indicator of healing in vivo. The wounds are irregular in outline (as depicted in Figure 5), thus a diameter, chosen at a random point on the wound circumference, may vary widely depending on the point chosen and is subject to skewing by where the investigator chooses to measure. Thus, this measurement does not represent "wound healing."

Recommend removing figure 5 and the in vivo data (perhaps moving to the supplement as supportive) and focusing the manuscript on the in vitro data. The manuscript should be edited to change any reference to "wound healing" to "wound healing in vitro" (as should the title be changed.)

Minor concerns:

Still needs English language editing. See for example: page 2 line 82

Please provide additional HPLC methodology, sufficient for other investigators to repeat this work independently using the exact same methods.